# BRIDGE - BUILDING REINFORCEMENT-LEARNING DEPTH-TO-IMAGE DATA GENERATION ENGINE FOR MONOCULAR DEPTH ESTIMATION

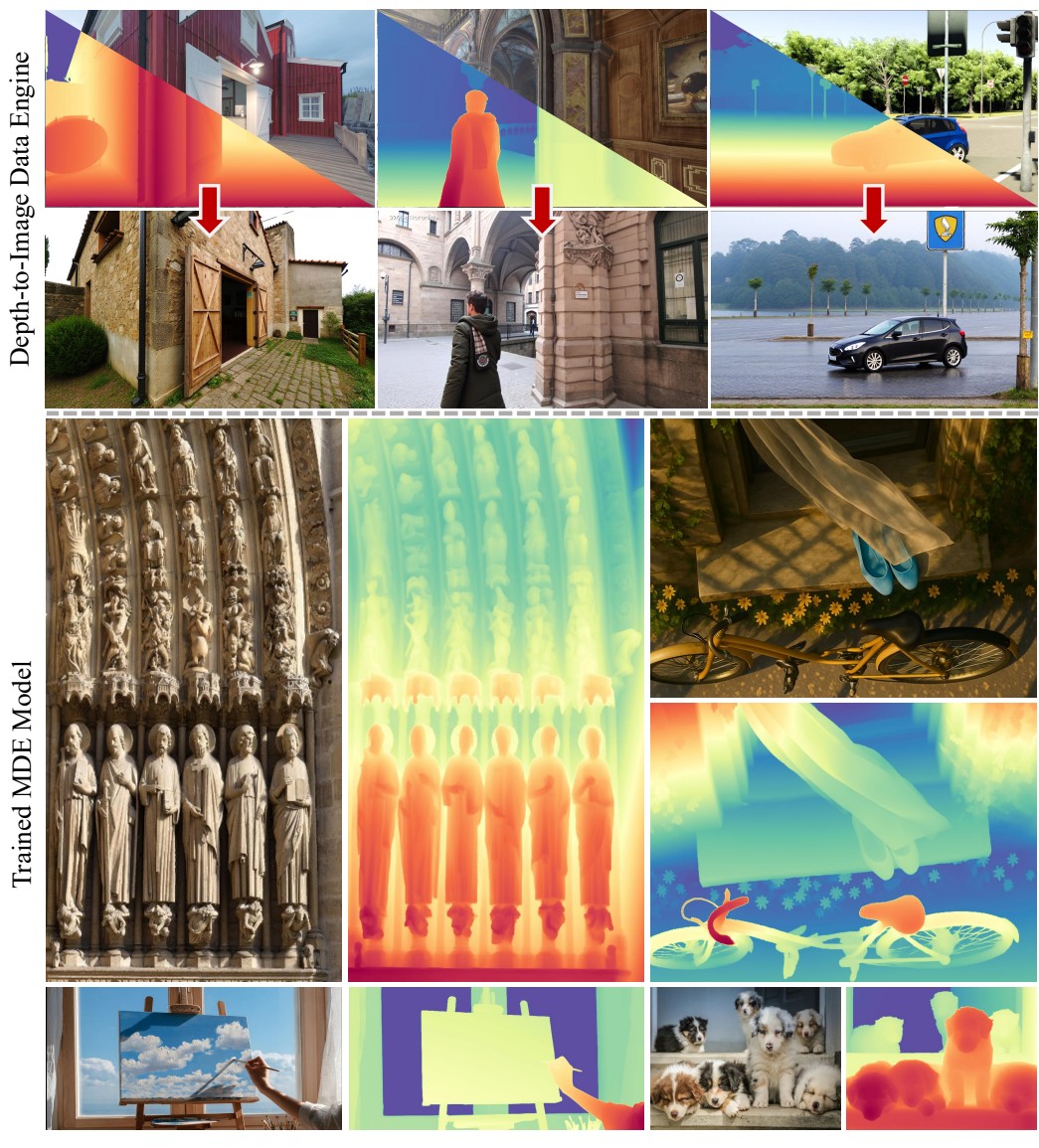

Figure 1: We present BRIDGE, showcasing its RL-optimized Depth-to-Image (D2I) data generation engine which is used for generating realistic and geometrically accurate RGB images from source depth maps and Monocular Depth Estimation (MDE) model which after being trained on the massive high-quality data generated by the D2I engine, achieves superior depth prediction in complex scenes.

## ABSTRACT

Monocular Depth Estimation (MDE) is a foundational task for computer vision. Traditional methods are limited by data scarcity and quality, hindering their robustness. To overcome this, we propose BRIDGE, an RL-optimized depth-to-image (D2I) generation framework that synthesizes over 20M realistic and geometrically accurate RGB images, each intrinsically paired with its ground truth depth, from diverse source depth maps. Then we train our depth estimation model on this dataset, employing a hybrid supervision strategy that integrates teacher pseudo-labels with ground truth depth for comprehensive and robust training. This innovative data generation and training paradigm enables BRIDGE to achieve breakthroughs in scale and domain diversity, consistently outperforming existing state-of-the-art approaches quantitatively and in complex scene detail capture, thereby fostering general and robust depth features. Code and models are available at `https://dingning-liu.github.io/bridge.github.io/`.

## 1 INTRODUCTION

Monocular Depth Estimation (MDE) stands as a cornerstone in the field of 3D computer vision, providing essential geometric perception for various critical applications such as 3D reconstruction (Mildenhall et al., 2021; Kerbl et al., 2023; Charatan et al., 2024; Ye et al., 2024), autonomous driving (Wang et al., 2019b), robotics (Wofk et al., 2019), and AR/VR (Rasla & Beyeler, 2022). The field has witnessed notable progress in recent years (Atapour-Abarghouei & Breckon, 2018; Ranftl et al., 2020; Bhat et al., 2023; Hu et al., 2024; Rajpal et al., 2023; Yang et al., 2024b; Ke et al., 2024; Yang et al., 2024c;d; Fu et al., 2024; Wang et al., 2025), however, training robust MDE models with excellent generalization remains challenging due to the scarcity of high-quality, precise ground truth depth annotations, insufficient detail and diversity in existing labels, and the underutilization of available depth data, creating a critical bottleneck for MDE model training.

Existing data acquisition and utilization methods can be categorized into three groups. (1) Bhat et al. (2023); Hu et al. (2024); Piccinelli et al. (2024) primarily rely on real-world data. While valuable, this approach is constrained by the sparsity of depth maps from sensors and the difficulty of accurately capturing and annotating transparent or reflective objects (Costanzino et al., 2023).

(2) Atapour-Abarghouei & Breckon (2018); Rajpal et al. (2023); Yang et al. (2024b); Ke et al. (2024) are mainly based on synthetic data. Although synthetic data is theoretically precise, it may introduce significant domain gaps (Ganin & Lempitsky, 2015) and geometric artifacts, degrading models' generalization capabilities.

(3) Ranftl et al. (2020); Yang et al. (2024c;d); Fu et al. (2024); Wang et al. (2025) integrate diverse data sources. MiDaS (Ranftl et al., 2020) as the pioneer significantly enhances generalization by training on various mixed datasets; however, its data coverage and diversity remain limited. Building upon this, Depth Anything (Yang et al., 2024c;d) advances SOTA by leveraging massive real-world images and a teacher model for pseudo-label generation, significantly improving generalization. However, its dependence on large real datasets and pseudo-label inaccuracies (despite high generation quality) bottleneck further performance and efficiency. These data-level limitations—diversity, scale, and pseudo-label fidelity—hinder truly universal and efficient MDE models.

To address the above challenges, we propose BRIDGE, an RL-optimized, large-scale Depth-to-Image (D2I) generation data engine. This engine leverages precise ground truth from synthetic depth datasets to generate massive, high-quality, and diverse RGB-D training data. Data created by the data engine enable MDE models to achieve breakthroughs in both scale and domain diversity, thereby fostering general, robust depth features and excellent real-world performance.

Specifically, our core lies in an innovative data generation pipeline: we first train an RL-optimized Depth-to-Image (D2I) generation model on many synthetic datasets. This model can directly utilize diverse source depth maps (from existing synthetic depth datasets) to synthesize over 20 million diverse and information-rich RGB images. Based on this, we introduce a hybrid data supervision strategy. This strategy pairs generated RGB images with original high-precision ground truth depth, which is screened through similarity detection methods such as SSIM and gradient analysis. Si-

multaneously, it leverages a large number of pseudo-labels generated by a teacher model trained on synthetic data, and both are jointly used for training. This ensures that the generated RGB images receive reliable supervision during training while improving the overall quality and utilization efficiency of the labels.

Our experimental results consistently demonstrate BRIDGE's superior performance across various challenging benchmarks, including indoor, outdoor, and synthetic animation environments, surpassing existing state-of-the-art methods. Qualitatively, BRIDGE excels at capturing fine-grained details and maintaining robustness in complex scene structures. Furthermore, ablation studies confirm the effectiveness of the RL-D2I generated data and the hybrid supervision training method.

Our main contributions are as follows:

- **An efficient data generation and utilization paradigm:** We adopt an RL-driven D2I paradigm, efficiently generating massive high-quality RGB-D data, which effectively alleviates data scarcity and quality issues.

- **Key practice of hybrid depth supervision training strategy:** We practice a hybrid depth supervision strategy, combining teacher model pseudo-labels with high-precision ground truth depth, to achieve a more challenging optimization objective and learn geometric structural knowledge from RGB data.

- **Superior performance and high training efficiency:** Our data generation and training strategy, using only approximately 20M data, surpasses SOTA models (e.g., Depth Anything V2 which uses 62M data), significantly improving training efficiency and demonstrating excellent detail capture capability and robustness.

## 2 RELATED WORKS

### 2.1 MONOCULAR DEPTH ESTIMATION

Monocular Depth Estimation (MDE), a foundational computer vision task, has seen significant progress with deep learning, moving beyond early methods limited by data homogeneity (Saxena et al., 2008). Deep learning approaches are broadly categorized into Discriminative Methods and Generative Methods. The former, starting with CNN-based pioneers like (Krizhevsky et al., 2009; Eigen et al., 2014; Eigen & Fergus, 2015; Chen et al., 2016) and later enhanced by multi-dataset training in MiDaS (Ranftl et al., 2020), learns a direct mapping from image to depth (Fu et al., 2018). The adoption of the Transformer architecture (Dosovitskiy et al., 2020; Ranftl et al., 2021; Liu et al., 2021) and large-scale self-supervised learning, as in Depth Anything (Yang et al., 2024c), further improves generalization, while models like NDDdepth (Shao et al., 2023a) and Depth Pro (Bochkovskii et al., 2024) target high-precision boundary estimation. In contrast, generative methods utilize models like Diffusion Models (Ho et al., 2020; Song et al., 2020; Nichol & Dhariwal, 2021; Ji et al., 2023) to synthesize depth maps, with works such as LeReS (Yin et al., 2021) and Metric3D v2 (Hu et al., 2024) focusing on accurate metric depth. The recent Depth Anything V2 (Yang et al., 2024d) combines large-scale synthetic images and pseudo-labeling to boost performance. Despite these advancements, a key challenge remains: achieving highly accurate and generalizable depth estimation in complex, unseen scenarios. To tackle this, we propose BRIDGE, a framework that generates a massive dataset of realistic and geometrically accurate RGB images to expand the effective training data.

### 2.2 SELF-SUPERVISED AND HYBRID SUPERVISION LEARNING

To train a robust MDE model despite the scarcity of precisely labeled data, researchers have explored various weakly supervised and self-supervised approaches, such as utilizing unlabeled monocular or stereo video data through photometric consistency loss for training (Godard et al., 2017; Kuznietsov et al., 2017; Ren et al., 2020). Other self-supervised approaches also include consistency training methods like FixMatch (Sohn et al., 2020) and unsupervised data augmentation (Xie et al., 2020), as well as strategies like shrinking class space (Yang et al., 2023a) to enhance certainty. Building on this, Pseudo-labeling supervision strategies (Lee et al., 2013), exemplified by Depth Anything V2 (Yang et al., 2024d), significantly improve model generalization by employing a powerful

teacher model to generate pseudo-labels for massive amounts of unlabeled images. However, despite the broad coverage provided by pseudo-labels, their inherent noise and inaccuracy—especially around boundaries and fine details—remain a bottleneck for further enhancing depth estimation performance. To leverage both the scale of pseudo-labels and the precision of ground truth annotations, we propose a hybrid supervision strategy. Unlike methods that solely rely on pseudo-labels, our strategy integrates high-precision ground truth depth from the original data via a similarity detection mechanism. By utilizing the teacher model's pseudo-labels for broad training and the similarity-guided ground truth for precision fine-tuning, our hybrid approach provides a more reliable and comprehensive supervision signal.

## 2.3 DEPTH-TO-IMAGE GENERATION AND SYNTHETIC DATA

The performance of Monocular Depth Estimation (MDE) is directly linked to the scale and quality of training data, yet acquiring high-quality ground truth depth from real-world data is expensive and lacks sufficient diversity, leading researchers to increasingly utilize synthetic data. Traditional synthetic methods, which rely on 3D rendering engines, provide geometrically accurate annotations but often suffer from a pronounced domain gap (Atapour-Abarghouei & Breckon, 2018; Rajpal et al., 2023) and low generation efficiency. More recently, methods based on Generative Models have been explored, such as using Diffusion Models (Ke et al., 2025; Gui et al., 2025) to generate depth maps from RGB images, but these primarily focus on depth prediction rather than the active generation of high-fidelity RGB-D data pairs for model training. To address this data bottleneck, Depth-to-Image (D2I) generation emerges as an active data augmentation strategy, synthesizing geometrically consistent RGB images from existing depth data. However, conventional D2I models often struggle with precise geometric alignment, leading to artifacts. For instance, Atlantis (Zhang et al., 2024) pioneers the use of a D2I approach but relies on triplet supervision primarily for domain-specific underwater scenes style transfer, rather than enforcing strict, geometric consistency across diverse domains. Our BRIDGE framework alleviates this by employing a Reinforcement Learning (RL) optimization mechanism in its D2I engine, ensuring both visual realism and geometric consistency for the massive generated dataset.

## 3 METHOD

To address data scarcity and label quality in monocular depth estimation, we propose a comprehensive three-stage pipeline, as illustrated in Figure 2. First, we train a powerful giant teacher model on large-scale synthetic data, and train a depth-to-image model using reinforcement learning. Second, we generate millions of visually realistic RGB images with accurate geometric structures, highly consistent with source depth maps, and their corresponding pseudo-labels, using the model we train in the previous stage (Section 3.1). Then we use a simple similarity algorithm method to calculate the original images and their corresponding generated images to create a mask for similar regions, and use the high-precision mask area and the original gt depth map for fine-tuning (Section 3.2). Finally, we train the MDE model on the large-scale generated data and their pseudo-labels (Section 3.3).

### 3.1 REINFORCEMENT LEARNING-OPTIMIZED DEPTH-TO-IMAGE GENERATION

To acquire large-scale, high-quality RGB-depth image pairs, we develop and train a reinforcement learning (RL)-optimized Depth-to-Image (D2I) generation model. This model generates visually realistic RGB images from source depth maps (from existing synthetic datasets like Hypersim (Roberts et al., 2021), TartanAir (Wang et al., 2020)) while precisely preserving their geometric structure. This generation process results in a large-scale dataset comprising approximately 20M images.

**Objectives and Advantages:** Leveraging the reinforcement learning paradigm, we ensure generated images are not only visually realistic but also geometrically accurate and consistent. This enables synthesizing diverse RGB images from a single depth map while preserving its true geometric structure. This approach mitigates geometric artifacts or structural distortions common in traditional D2I models and expands the diversity of effective training data.

**Training Strategy:** As shown in Figure 3, our D2I model, inspired by VADER (Prabhudesai et al., 2024), is trained via reward-gradient-driven direct optimization to efficiently and accurately generate

Figure 2: **BRIDGE Pipeline.** First training a depth-to-image model to synthesize millions of realistic RGB images with precise ground truth depths and a teacher model for pseudo labeling. Student model is then trained on this extensive synthetic dataset. Finally, it's fine-tuned using mask-based refinement with original ground truth depth for robust generalization and detailed depth capture.

high-fidelity images. This direct usage of reward gradients guides the generation process towards desired visual quality, thereby avoiding complex proxy objective functions and improving training memory efficiency. The overall optimization objective minimizes the following loss:

$$L_{\text{total}}(\theta) = \lambda_{\text{depth}} \cdot L_{\text{D}}(\mathcal{M}(D_{\text{source}}, \theta), D_{\text{source}}) - \lambda_{\text{aesthetic}} \cdot R_{\text{aesthetic}}(\mathcal{M}(D_{\text{source}}, \theta)), \quad (1)$$

where $D_{\text{source}}$ represents the input source depth map, $\mathcal{M}(D_{\text{source}}, \theta)$ denotes the RGB image generated by the D2I model $\mathcal{M}$ with parameters $\theta$ based on $D_{\text{source}}$, $L_{\text{D}}(D_{\text{gen}}, D_{\text{source}})$ is a cosine similarity loss. Its corresponding loss (defined as $1 -$ cosine similarity reward) aims to minimize the difference between the depth map $D_{\text{gen}}$ inverted from the generated RGB image and the source depth map $D_{\text{source}}$. This ensures the D2I model precisely learns and maintains the scene's geometric structural information. $R_{\text{aesthetic}}(\mathcal{M}(D_{\text{source}}, \theta))$ is the aesthetic reward function, quantifying the visual quality and aesthetic appeal of the generated image. It is formulated as:

$$R_{\text{aesthetic}}(I) = g_{\text{MLP}}\left(\frac{f_{\text{CLIP}}(I)}{\|f_{\text{CLIP}}(I)\|_2}\right), \quad (2)$$

where $f_{\text{CLIP}}(I)$ represents the high-dimensional feature embedding extracted from the input image $I$ by a pre-trained CLIP image encoder. $\|\cdot\|_2$ denotes the L2 norm, normalizing the extracted features to unit length to prevent their magnitude from unduly influencing subsequent scoring and to align with the MLP's training process. Finally, $g_{\text{MLP}}(\cdot)$ is a specialized MLP head that receives the normalized CLIP features as input and maps them to a scalar aesthetic score $S_{\text{aesthetic}} \in \mathbb{R}$. During D2I model optimization, we maximize the aesthetic quality of generated images by minimizing the aesthetic loss term $L_{\text{aesthetic}}(I) = -R_{\text{aesthetic}}(I)$, thereby incentivizing the model to produce images with higher aesthetic scores.

The model is trained using gradient descent, with the parameter update rule:

$$\theta \leftarrow \theta - \eta \cdot \nabla_\theta L_{\text{total}}(\theta), \quad (3)$$

where $\eta$ is the learning rate. The core challenge is computing the gradient of the loss function with respect to model parameters $\theta$, $\nabla_\theta L_{\text{total}}(\theta)$. Since $x_0$ is the final output of the diffusion model's multi-step denoising process, and model parameters $\theta$ are involved in each denoising prediction $\epsilon_\theta(x_t, t, D_{\text{source}})$, the gradient must be backpropagated through the entire reverse diffusion process. Specifically, the gradient of the loss with respect to parameters can be expressed as:

$$\nabla_\theta L_{\text{total}}(\theta) = \sum_{t=0}^{T} \frac{\partial L_{\text{total}}(x_0)}{\partial x_t} \cdot \frac{\partial x_t}{\partial \theta}, \quad (4)$$

where $\frac{\partial L_{\text{total}}(x_0)}{\partial x_t}$ represents the sensitivity of the loss to the intermediate diffusion step $x_t$, and $\frac{\partial x_t}{\partial \theta}$ denotes the influence of model parameters on the intermediate state. This gradient propagation efficiently guides the diffusion model to generate high-quality RGB images while precisely capturing and maintaining the geometric structure of the input depth map, significantly enhancing training sample and computational efficiency.

Figure 3: **Reward Model Process.** Our D2I model is trained via reward-gradient-driven direct optimization , avoiding complex proxy objective functions and improving training memory efficiency.

## 3.2 DEPTH PSEUDO-LABEL GENERATION AND MULTI-SOURCE DEPTH FUSION STRATEGY

After generating millions of high-fidelity RGB images, we construct their corresponding depth labels using a multi-strategy fusion approach. These are then combined with the original synthetic data to form the final hybrid training set.

**Teacher Model-Based Initial Pseudo-Label Generation:** We leverage a powerful teacher model trained on 1M synthetic data to infer initial relative depth pseudo-labels for all 20 million generated RGB images. These pseudo-labels provide valuable, geometrically consistent supervision, crucial for bridging the domain gap.

**Similarity-Guided Ground Truth Depth Utilization:** To achieve more precise and reliable depth labels and maximize the use of all available depth information (including original high-precision ground truth depth), we introduce a similarity-based method. This addresses pixel-level inaccuracies or detail loss from relying solely on teacher-generated pseudo-labels. Specifically, we construct a fusion mask for each generated RGB image $I_{gen}$ by evaluating its similarity to the original synthetic RGB image $I_{orig}$. This mask generation involves two main steps:

- **Feature-based Registration for Structural Similarity:** First, we geometrically register the generated RGB image ($I_{gen}$) with its original synthetic counterpart ($I_{orig}$) using ORB feature detection and matching (Karami et al., 2017). This step compensates for minor shifts or deformations during image generation, aligning them at the pixel level. After registration, we compute the Structural Similarity Index Measure (SSIM) map (Nilsson & Akenine-Möller, 2020) between the aligned images. Regions with an SSIM value above a predefined threshold are identified as high-similarity areas, forming the first binary mask.

- **Direct SSIM-based Pixel-level Similarity:** Second, we directly compute the SSIM map between the size-adjusted generated RGB image ($I_{gen}$) and the original synthetic RGB image ($I_{orig}$). This evaluates their pixel-level similarity without explicit geometric registration. Regions where the SSIM value exceeds the same threshold form the second binary mask.

The final fusion mask is generated by applying a logical OR operation to these two binary masks. This ensures that any high-similarity region identified by either method is included. To enhance mask robustness and reduce noise, we further apply morphological opening and closing operations to smooth boundaries and fill small gaps.

Upon obtaining the final fusion mask, we integrate this refined depth supervision into a two-stage training process. Initially, the BRIDGE model undergoes pre-training on a large-scale dataset, primarily using the teacher-generated pseudo-labels. Following this, a fine-tuning stage is performed. In this fine-tuning stage, for regions covered by the fusion mask, we directly utilize the corresponding high-precision ground truth depth ($D_{gt}$) from the original synthetic data as training labels. This strategy ensures the model first learns broad geometric consistency from vast pseudo-labeled data and then refines its accuracy and detail with the most accurate supervision in high-precision, geometrically consistent areas, thereby significantly improving depth estimation accuracy and detail.

**Final Hybrid Dataset Composition:** The composite dataset for training BRIDGE comprises two main components. The first consists of large-scale generated RGB images: approximately 20 million high-fidelity RGB images produced by the RL-optimized D2I process, each paired with its teacher-model-derived depth pseudo-label. The second involves original synthetic data depth labels: we

directly integrate depth labels from multiple original synthetic datasets, crucially including the high-precision ground truth depth filtered by our similarity-guided strategy.

### 3.3 Training Monocular Depth Estimation Model

After designing the data generation strategy, we train the Monocular Depth Estimation (MDE) model on the generated data. For model architecture, we adopt the established pretrained DINOv2-Giant encoder (Oquab et al., 2023) and DPT (Ranftl et al., 2021) head combination from Depth Anything (Yang et al., 2024c). To optimize our model, we utilize two loss terms, both of which are based on the approach proposed in MiDaS(Ranftl et al., 2020) because they are well-suited for affine-invariant depth. The first is the scale- and shift-invariant loss ($L_{ssi}$), which ensures the model's robustness to variations in the scale and offset of depth values. The second is the gradient matching loss ($L_{gm}$), which helps the model capture fine-grained scene structure and details by enforcing similarity between the gradients of the predicted and ground-truth depth maps. We set the $L_{ssi}$ and $L_{gm}$ as 1:4 and we ignore its top n largest loss regions during training, where n is set as 10% because the regions are usually considered as potentially noisy pseudo labels. To enable zero-shot metric depth estimation for arbitrary "in-the-wild" images, we further introduce a specialized scale head. This head autonomously predicts the image's metric scale, significantly enhancing the model's accuracy and generalization in real-world scenarios. We first fine-tune the metric depth model, then decouple and train the Scale Head separately. This improves training efficiency and provides high-quality, artifact-free, precise metric depth for downstream tasks like novel view synthesis, further enhancing the model's utility.

## 4 Implementation Details

Our RL-D2I generative model is based on the architecture of Flux.1 Depth[dev] (Labs, 2024) and fine-tuned using depth data from synthetic datasets, and the learning rate is 1e-5. The generator synthesizes approximately 20 million visually realistic and geometrically accurate RGB images from source depth maps by optimizing a loss function that includes depth loss (weight 0.9) and aesthetic reward (weight 0.1). To achieve high geometric accuracy and depth similarity, we ensure that the generated images meet stringent quality criteria: high-precision regions are identified by combining areas where: 1) ORB feature-based registered SSIM between the generated and original RGB images exceeds 0.85 (requiring at least 10 ORB matches), and 2) direct SSIM between them also exceeds 0.85. We only select mask samples where the valid region constitutes over 50% of the pixels, and a 3x3 erosion operation is performed to filter out excessively small regions. Subsequently, a 518x518 crop is randomly extracted, centered on the largest valid region's bounding box. For each original depth map, we generate four RGB images simultaneously, and use different random seeds. The prompt is set to None during the generation process.

## 5 Experiments

In this section, we will validate the effectiveness and advantages of the proposed method for monocular depth estimation through a series of experiments. We first introduce the datasets, evaluation metrics, and experimental settings. Then, we present the performance of our method on multiple benchmark datasets and conduct ablation experiments to analyze the contribution of each module to the overall performance.

### 5.1 Evaluation Datasets

For monocular depth estimation, we conduct a series of experiments to evaluate the performance of our model on five widely used benchmarks. NYUv2 (Silberman et al., 2012) and ScanNet (Dai et al., 2017) provide indoor RGB-D data captured by depth sensors. ETH3D (Schops et al., 2017) includes both indoor and outdoor scenes, with depth data collected via laser scanners. KITTI (Geiger et al., 2012) consists of outdoor driving scenes captured with cameras and LiDAR sensors. For ScanNet, we extract 1400 test frames. Sintel (Butler et al., 2012) is a synthetic dataset derived from animated short films, from which we extract images of 600 training samples for evaluation.

Table 1: **Quantitative Comparison with SOTA Methods on Zero-Shot Relative Depth Estimation.** These benchmarks aim to comprehensively measure model generalization ability and accuracy in diverse environments. While these standard benchmarks provide a direct quantitative comparison, some specific strengths of a model, such as details, complex layouts, may not be fully reflected by these benchmarks.The underline represents the best performance of SOTA models on this dataset.

| | Method | KITTI | | NYUv2 | | ScanNet | | ETH3D | | Sintel | | DA2K |
|---|---|---|---|---|---|---|---|---|---|---|---|---|
| | | $\delta_1 \uparrow$ | AbsRel↓ | $\delta_1 \uparrow$ | AbsRel↓ | $\delta_1 \uparrow$ | AbsRel↓ | $\delta_1 \uparrow$ | AbsRel↓ | $\delta_1 \uparrow$ | AbsRel↓ | Acc(%) |
| *Generative* | DepthFM (Gui et al., 2025) | 0.932 | 0.085 | 0.956 | 0.065 | 0.947 | 0.068 | 0.958 | 0.069 | 0.700 | 0.577 | 85.8 |
| | Marigold (Ke et al., 2024) | 0.916 | 0.099 | 0.964 | 0.055 | 0.951 | 0.064 | 0.960 | 0.065 | 0.703 | 0.576 | 86.8 |
| | GeoWizard (Fu et al., 2024) | 0.921 | 0.097 | 0.966 | 0.052 | 0.953 | 0.061 | 0.961 | 0.064 | 0.705 | 0.574 | 88.1 |
| *Discriminative* | LeReS (Yin et al., 2021) | 0.784 | 0.149 | 0.916 | 0.090 | 0.917 | 0.091 | 0.777 | 0.171 | 0.590 | 0.785 | - |
| | DPT (Ranftl et al., 2021) | 0.881 | 0.111 | 0.919 | 0.091 | 0.932 | 0.084 | 0.929 | 0.115 | 0.605 | 0.700 | - |
| | MiDaSv3.1 (Birkl et al., 2023) | 0.851 | 0.126 | 0.980 | 0.048 | 0.957 | 0.069 | 0.947 | 0.078 | 0.670 | 0.611 | - |
| | Metric3Dv2 (Hu et al., 2024) | 0.968 | 0.062 | 0.963 | 0.058 | 0.941 | 0.074 | 0.960 | 0.066 | 0.662 | 0.619 | - |
| | Depth Anything V2 (Yang et al., 2024d) | 0.946 | 0.075 | 0.979 | 0.045 | 0.978 | 0.043 | 0.988 | 0.038 | 0.672 | 0.598 | 97.1 |
| | Depth Pro (Bochkovskii et al., 2024) | 0.498 | 0.375 | 0.580 | 0.245 | 0.667 | 0.207 | 0.687 | 0.201 | 0.476 | 0.883 | - |
| | **Ours** | **0.938** | **0.081** | **0.982** | **0.041** | **0.981** | **0.033** | **0.991** | **0.029** | **0.719** | **0.513** | **97.3** |

## 5.2 EVALUATION METRICS

All evaluations are carried out under the zero-shot configuration. Following previous works (Yang et al., 2024d; He et al., 2025), the evaluation of affine-invariant inverse depth prediction is performed by optimizing the scale and shift discrepancies between the estimated depth and the ground truth. For quantitative analysis, we utilize the absolute relative error (AbsRel) and $\delta 1$ accuracy. AbsRel is calculated as $\frac{1}{N} \sum_{k=0}^{N-1} \frac{|\hat{x}_d - x_d|}{x_d}$, where $N$ represents the total pixel count. $\delta 1$ accuracy measures the percentage of pixels where the maximum ratio between the predicted affine-invariant inverse depth and the inverse true depth falls below $1.25$.

## 5.3 ZERO-SHOT DEPTH ESTIMATION

Our model demonstrates exceptional zero-shot generalization in depth estimation across both indoor and outdoor datasets, highlighting its robustness, fine-grained detail, and consistent depth estimation for objects.

**Quantitative Comparisons:** Table 1 shows the results that our method achieves outstanding performance across multiple datasets. Our method achieves outstanding performance across multiple datasets and establishes new state-of-the-art (SOTA) results on several mainstream benchmarks. Specifically, our model demonstrates overwhelming superiority on indoor scene datasets like NYUv2, ScanNet, and ETH3D, with its ability to generate fine-structure predictions aligning perfectly with its objectives. Furthermore, our model slightly surpasses Depth Anything V2 on benchmarks like DA2K, further confirming its excellent zero-shot generalization capabilities. Notably, Depth Pro's significantly lower performance in this evaluation is related to its primary optimization for metric depth rather than relative depth. Lastly, we do not achieve optimal performance on KITTI, primarily because of the dataset's inherent sparsity. Our model is designed to capture fine-grained global and local depth information, which is not fully reflected in the KITTI evaluation.

**Qualitative Comparisons:** Figure 4 presents qualitative monocular depth estimation results, highlighting our model's superior capability to capture fine-grained details and robustly handle challenging objects compared to Depth Anything V2, Depth Pro, and generative methods. Notably, our model successfully identifies and accurately estimates the depth of reflective surfaces, such as mirrors, on the NYUv2 dataset. It also preserves complex details, like distant table legs in ScanNet, and captures the fine textures of remote architectural and wooden structures on the Sintel dataset. Furthermore, it effectively processes similarly colored objects; for instance, on the KITTI dataset, it precisely delineates a person's head from the background, a nuance Depth Anything V2 often struggles with. Crucially, our model exhibits excellent generalization to "in-the-wild" data, exem-

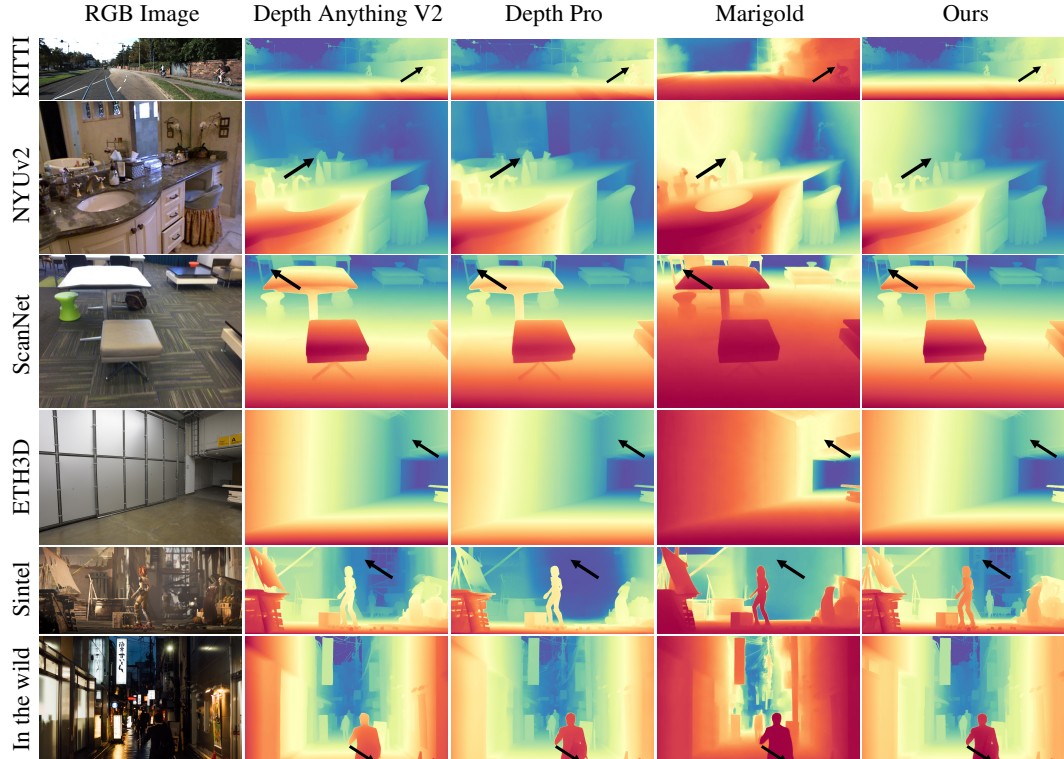

Figure 4: **Qualitative comparison of relative depth estimation across different datasets.** Our model captures fine-grained details and generalizes robustly on in-the-wild data.

plified by its accurate depth estimation for transparent umbrellas, fully demonstrating its robustness in complex real-world scenarios.

## 5.4 FINE-TUNED TO METRIC DEPTH ESTIMATION

To validate our model's generalization ability in metric depth estimation, we also transfer our pretrained encoder and design an additional scale head to adapt to both indoor and outdoor domains. The scale head is designed to validate the model's generalization ability in metric depth estimation, adapting to both indoor and outdoor domains. Structurally, it uses the DINOv2 encoder and an MLP head to predict an accurate metric scale. Following the pipeline of Depth Anything V2, our model maintains a consistently strong performance level on both NYUv2 and KITTI datasets shown in Table 4 and Table 5.

## 5.5 ABLATION STUDIES

In this section, we systematically evaluate the contribution of each factor in BRIDGE to the overall performance, aiming to quantify the impact of our proposed innovative data generation scheme. Our ablation experiments are conducted under consistent training configurations to ensure fair comparisons.

**Impact of BRIDGE Generated Data:** We first investigate the effectiveness of our RL-D2I generation engine, which serves as the core mechanism for alleviating data scarcity and enhancing training data diversity. To demonstrate its significant impact, we compare a baseline model using DINOv2 Base trained on 1M synthetic data from various large-scale synthetic depth datasets with the BRIDGE model generated 1M data (comprising 0.95M pseudo-labels and 0.05M masked original ground truth depth). As shown in Table 2, the integration of RL-D2I-generated data significantly boosts the performance across all evaluated metrics. This improvement underscores the effectiveness of our RL-D2I engine in generating visually realistic and geometrically accurate images, which effectively expands the scale and diversity of the training data.

Table 2: **Impact of Generated Data.** We demonstrate the effectiveness of our RL-D2I engine by comparing a model trained on standard synthetic data with one trained on 1M generated data with ViT-L. The results show that our generated data boosts performance across all metrics.

| Method | KITTI | | NYUv2 | | ScanNet | | ETH3D | | Sintel | |
|---|---|---|---|---|---|---|---|---|---|---|
| | $\delta_1 \uparrow$ | AbsRel$\downarrow$ | $\delta_1 \uparrow$ | AbsRel$\downarrow$ | $\delta_1 \uparrow$ | AbsRel$\downarrow$ | $\delta_1 \uparrow$ | AbsRel$\downarrow$ | $\delta_1 \uparrow$ | AbsRel$\downarrow$ |
| Baseline | 0.921 | 0.097 | 0.960 | 0.060 | 0.964 | 0.057 | 0.969 | 0.060 | 0.599 | 0.709 |
| **w/ Generated Data** | **0.929** | **0.092** | **0.964** | **0.056** | **0.968** | **0.048** | **0.970** | **0.059** | **0.612** | **0.692** |

Table 3: **Ablation Study of the Hybrid Training Strategy.** We compares the performance of BRIDGE's ViT-L model, trained solely on pseudo-labels with hybrid supervision integrating similarity-guided high-precision ground truth depth on benchmarks. The results highlight the importance of incorporating original ground truth depth for improving depth estimation accuracy and detail capture.

| Method | KITTI | | NYUv2 | | ScanNet | | ETH3D | | Sintel | |
|---|---|---|---|---|---|---|---|---|---|---|
| | $\delta_1 \uparrow$ | AbsRel$\downarrow$ | $\delta_1 \uparrow$ | AbsRel$\downarrow$ | $\delta_1 \uparrow$ | AbsRel$\downarrow$ | $\delta_1 \uparrow$ | AbsRel$\downarrow$ | $\delta_1 \uparrow$ | AbsRel$\downarrow$ |
| Pseudo-labels Only | 0.924 | 0.099 | 0.958 | 0.069 | 0.957 | 0.069 | 0.949 | 0.078 | 0.595 | 0.751 |
| **Full Hybrid Strategy** | **0.926** | **0.098** | **0.960** | **0.067** | **0.960** | **0.061** | **0.953** | **0.073** | **0.600** | **0.725** |

Table 4: The metric results on NYUv2.

| Method | Higher $\uparrow$ | Lower $\downarrow$ | |
|---|---|---|---|
| | $\delta_1$ | AbsRel | RMSE |
| AdaBins (Bhat et al., 2021) | 0.903 | 0.103 | 0.364 |
| DPT (Ranftl et al., 2021) | 0.904 | 0.110 | 0.357 |
| P3Depth (Patil et al., 2022) | 0.898 | 0.104 | 0.356 |
| SwinV2 (Liu et al., 2022) | 0.949 | 0.083 | 0.287 |
| IEBins (Shao et al., 2023b) | 0.936 | 0.087 | 0.314 |
| ZoeDepth (Bhat et al., 2023) | 0.951 | 0.077 | 0.282 |
| Metric3Dv2 (Hu et al., 2024) | 0.987 | 0.046 | 0.181 |
| Ours | **0.986** | **0.052** | **0.197** |

Table 5: The metric results on KITTI.

| Method | Higher $\uparrow$ | Lower $\downarrow$ | |
|---|---|---|---|
| | $\delta_1$ | AbsRel | RMSE |
| AdaBins (Bhat et al., 2021) | 0.964 | 0.058 | 2.360 |
| P3Depth (Patil et al., 2022) | 0.953 | 0.071 | 2.842 |
| SwinV2 (Liu et al., 2022) | 0.977 | 0.050 | 1.966 |
| GEDepth (Yang et al., 2023b) | 0.976 | 0.048 | 2.044 |
| IEBins (Shao et al., 2023b) | 0.978 | 0.050 | 2.011 |
| ZoeDepth (Bhat et al., 2023) | 0.971 | 0.054 | 2.281 |
| Metric3Dv2 (Hu et al., 2024) | 0.985 | 0.044 | 1.993 |
| Ours | **0.983** | **0.045** | **1.862** |

**Efficiency of Hybrid Training Strategy:** As shown in Table 3, our hybrid depth supervision strategy is crucial for combining the broad coverage of teacher model pseudo-labels with the precision of similarity-guided high-precision ground truth depth. This approach aims to provide more robust and reliable supervision signals for learning geometric structural knowledge. We conduct an ablation study to quantify the benefits of this dual supervision. We also test on 1M data, the BRIDGE model trained with our full hybrid strategy yields superior results compared to a variant that relies solely on teacher-generated pseudo-labels for the generated RGB images.

## 6 CONCLUSION

In this work, we introduce BRIDGE, an innovative Monocular Depth Estimation framework that addresses data scarcity and quality issues. This is achieved by leveraging an RL-optimized Depth-to-Image (RL-D2I) engine to generate over 20M RGB-D data, which, along with a hybrid strategy of teacher pseudo-labels and high-precision ground truth, achieves state-of-the-art performance, consistently outperforming methods like Depth Anything V2 on benchmarks with significantly less training data (20M vs. 62M). Our findings highlight BRIDGE's superior capability in capturing fine-grained details, ensuring geometric consistency, and demonstrating robust zero-shot generalization to complex in-the-wild scenes, paving the way for more efficient and generalizable MDE solutions.

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

# A EXPERIMENTS

## A.1 MODEL SELECTION

We compare the learning capabilities of different DINOv2 (Oquab et al., 2023) versions on our hybrid data. For DINOv2-Giant and Large, we select the non-register version as their representative. As shown in Table 6, we find that the Giant and Large versions of DINOv2 exhibit excellent learning ability on synthetic data and generalize remarkably well to real-world test datasets. In contrast, the Small and Base versions do not achieve satisfactory results. This might be attributed to their limited model capacity, which is insufficient to capture the rich fine-grained information and diversity present in generation data, thus leading to difficulties in generalizing to more complex real-world scenarios. Therefore, we ultimately choose to adopt the DINOv2-Giant for our model, to fully leverage its powerful generalization capabilities and accuracy.

| Method | KITTI | | NYUv2 | | ScanNet | | ETH3D | | Sintel | |
|---|---|---|---|---|---|---|---|---|---|---|
| | $\delta_1 \uparrow$ | AbsRel$\downarrow$ | $\delta_1 \uparrow$ | AbsRel$\downarrow$ | $\delta_1 \uparrow$ | AbsRel$\downarrow$ | $\delta_1 \uparrow$ | AbsRel$\downarrow$ | $\delta_1 \uparrow$ | AbsRel$\downarrow$ |
| DINOv2-S | 0.928 | 0.082 | 0.964 | 0.062 | 0.921 | 0.085 | 0.978 | 0.045 | 0.637 | 0.684 |
| DINOv2-B | 0.932 | 0.081 | 0.968 | 0.054 | 0.941 | 0.065 | 0.980 | 0.039 | 0.647 | 0.644 |
| DINOv2-L | 0.937 | 0.081 | 0.978 | 0.045 | 0.972 | 0.049 | 0.989 | 0.032 | 0.688 | 0.588 |
| DINOv2-G | **0.938** | **0.081** | **0.982** | **0.041** | **0.981** | **0.033** | **0.991** | **0.029** | **0.719** | **0.513** |

Table 6: Comparison among various size pre-trained DINOv2 encoders trained on our dataset.

Table 7: **Our training sources. * represents the quantity sampled from the dataset.**

| Dataset | Indoor | Outdoor | # Images |
|---|---|---|---|
| **Precise Synthetic Images (1M)** | | | |
| BlendedMVS (Yao et al., 2020) | ✓ | ✓ | 115K |
| TartanAir (Wang et al., 2020) | ✓ | ✓ | 306K |
| Hypersim (Roberts et al., 2021) | ✓ | | 60K |
| IRS (Wang et al., 2019a) | ✓ | | 103K |
| VKITTI 2 (Cabon et al., 2020) | | ✓ | 20K |
| Spring (Mehl et al., 2023) | | ✓ | 5K |
| DA-V (Yang et al., 2024a) | ✓ | ✓ | 400K* |
| **Generated Images (20M)** | | | |
| BlendedMVS | | ✓ | 2.2M |
| TartanAir | | ✓ | 2M |
| Hypersim | ✓ | ✓ | 2.4M |
| IRS | ✓ | ✓ | 4.2M |
| VKITTI 2 | | ✓ | 1M |
| Spring | | ✓ | 20K |
| DA-V | ✓ | ✓ | 8.2M |

## A.2 ADDITIONAL QUALITATIVE COMPARISON

**Training Datasets:** As shown in Table 7, we train both a generative model and a teacher model using seven synthetic datasets to enhance label precision. To effectively mitigate the limitations of synthetic images regarding distribution shift and limited diversity, we generate 20M synthetic data based on the aforementioned datasets, thereby significantly broadening the diversity of the overall samples. Notably, for datasets such as Hypersim, where the original data primarily covers indoor

scenes, our generation strategy successfully produces outdoor data largely consistent with depth maps, effectively expanding the scene coverage of the training samples.

**Qualitative Results:** Figure 5 and Figure 6 show additional qualitative comparisons with other state-of-the-art monocular depth estimation methods. For indoor, outdoor, and non-real scenes alike, our model consistently produces depth maps with higher fidelity than the other methods. Our method particularly excels in outdoor scenes, showing clear distinctions between distant objects and the sky, while Depth Pro (Bochkovskii et al., 2024) and Marigold (Ke et al., 2025) fail to distinguish objects that are farther away.

**Conditional Synthesis:** Figure 7 shows a comparison of depth-conditioned synthesis results. We first infer depth based on an image using our BRIDGE and the Depth Anything V2 model, and then use the most basic pre-trained depth-to-image ControlNet (Zhang, Rao, and Agrawala 2023) with Stable Diffusion 1.5 to synthesize new samples based on the depth maps and a text prompt. We can clearly observe that the depth maps obtained with Depth Anything V2 do not reflect the actual depth well and are additionally inaccurate for some parts of the image. In contrast, our method yields sharp and realistic depth maps. This result is also reflected in the synthesized results, where images created based on our depth map more closely resemble the actual image.

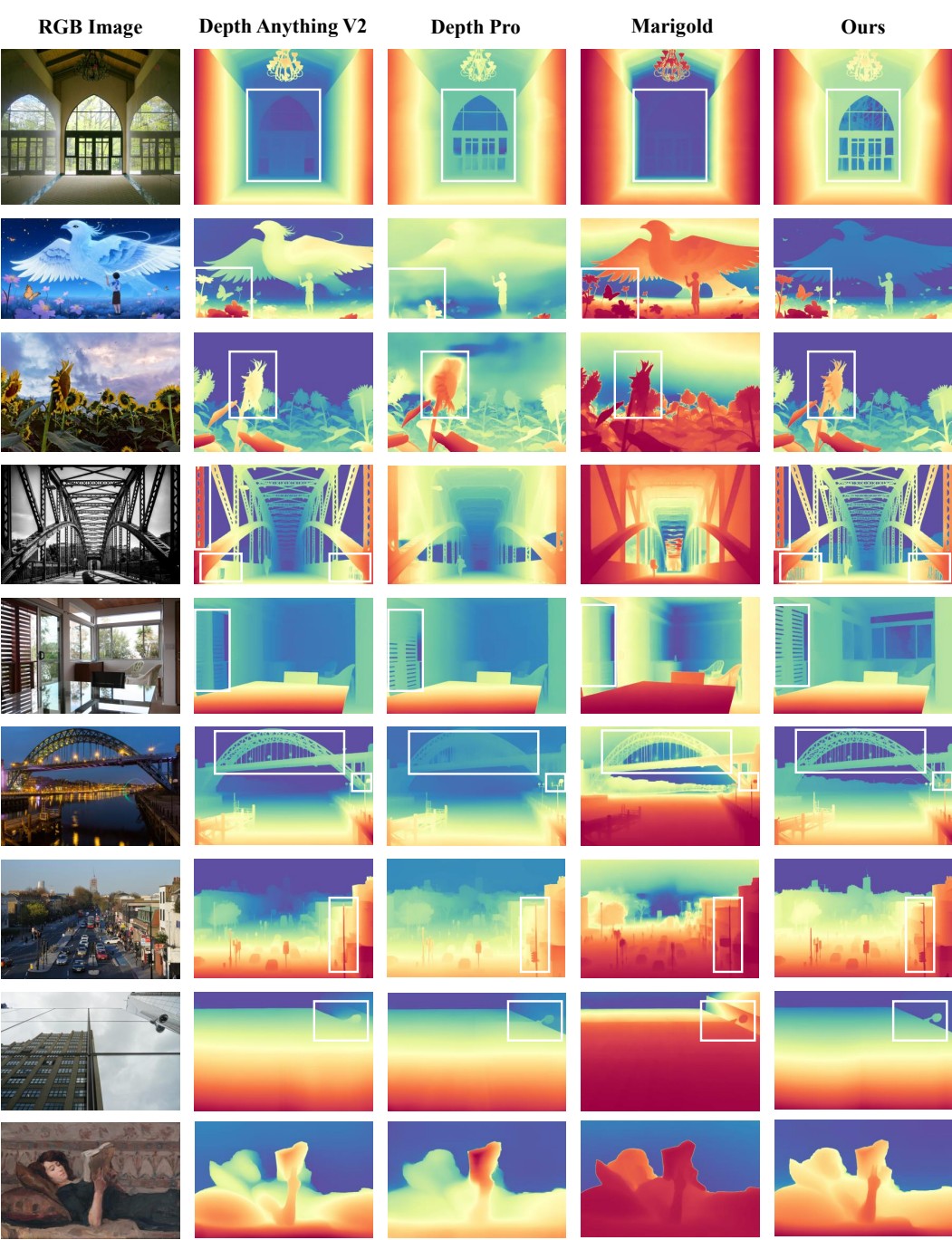

Figure 5: Comparison between our model and leading methods, including Depth Anything V2 (Yang et al., 2024d), Depth Pro(Bochkovskii et al., 2024) and Marigold(Ke et al., 2025) on open-world images.

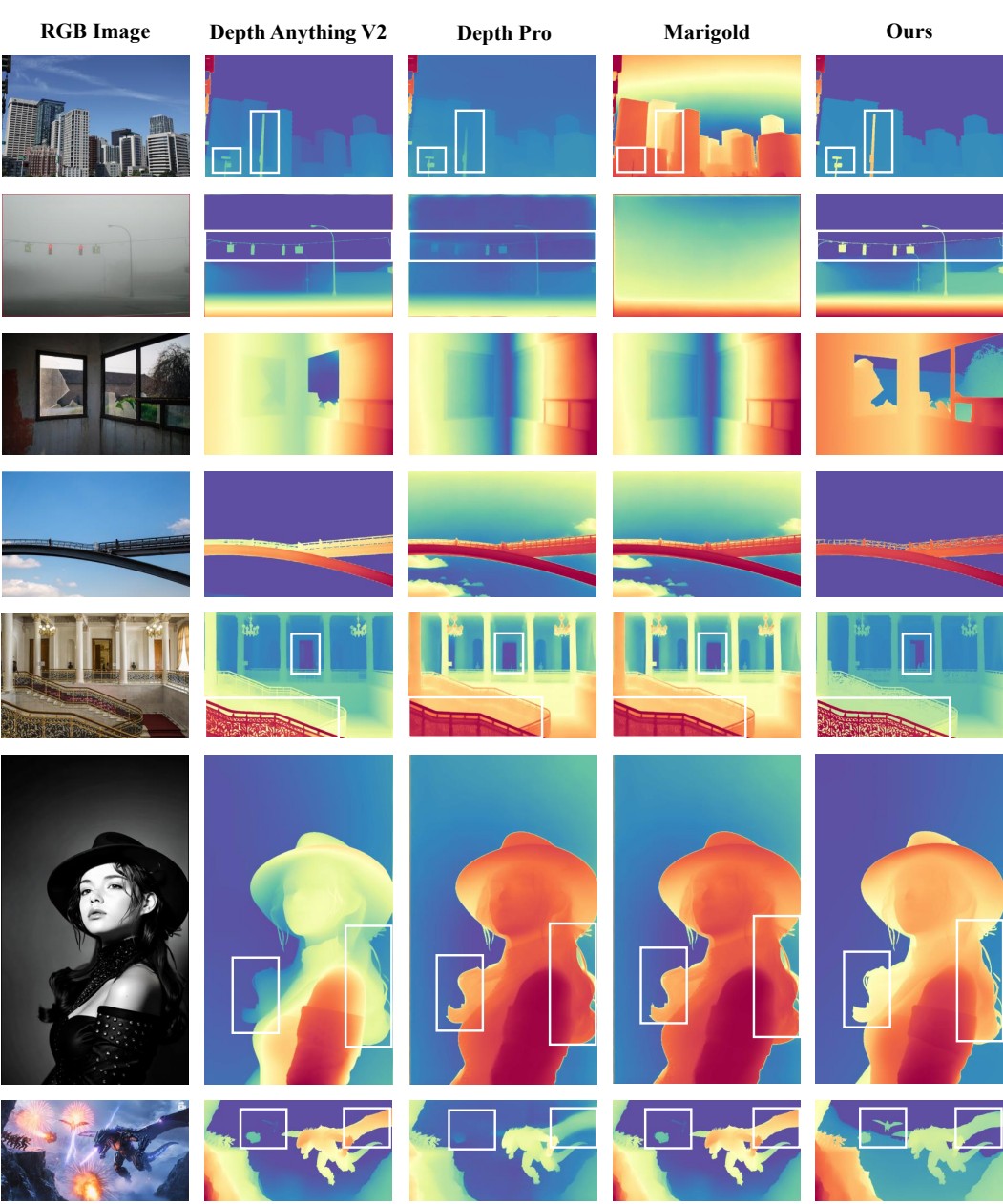

Figure 6: Additional comparison between our model and leading methods, including Depth Anything V2 (Yang et al., 2024d), Depth Pro(Bochkovskii et al., 2024) and Marigold(Ke et al., 2025) on "in-the-wild" images.

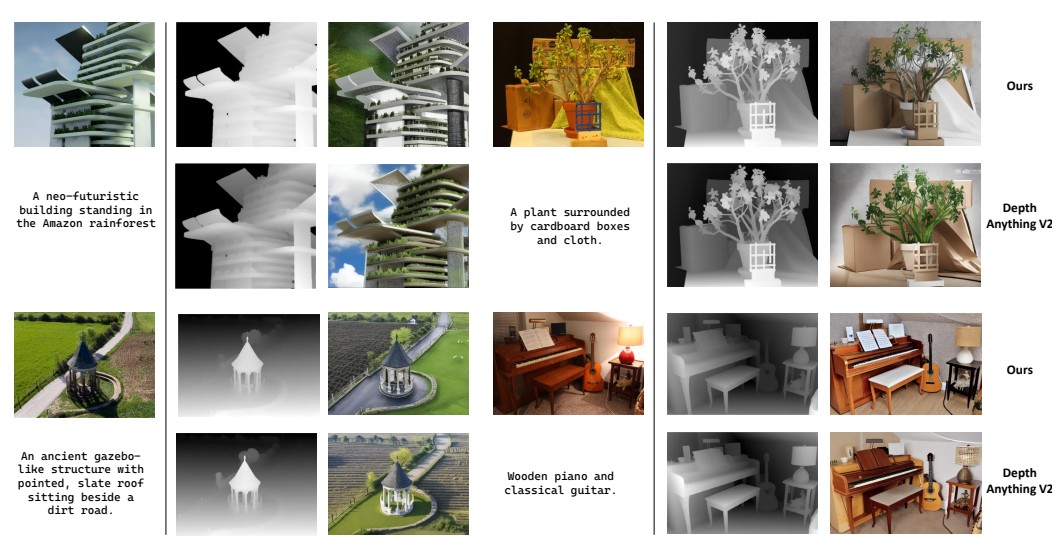

Figure 7: Our model, BRIDGE, generates superior, high-fidelity depth maps that enable ControlNet (Zhang et al., 2023) to synthesize new images with a zero-shot capability, precisely replicating the depth field of the source image. In contrast, Depth Anything V2 (Yang et al., 2024d) struggles to produce an accurate depth field, as demonstrated by the clear discrepancies between its corresponding ControlNet output and the source images. The prompts used for ControlNet are displayed in the lower left corners, and all images were generated with the same random seed.

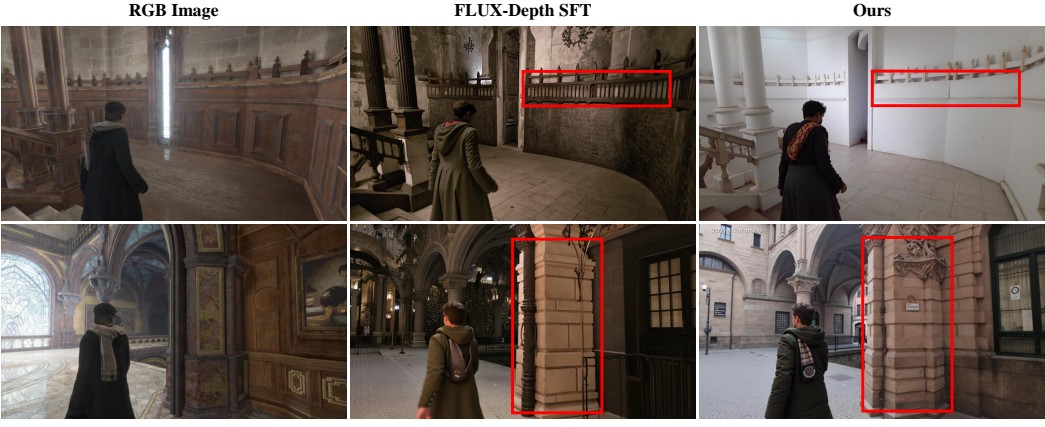

Figure 8: **Comparison between Standard SFT and RL-optimized fine-tuning.** We show sample generated images. Our RL-optimized variants enable the model to capture finer details than the Standard SFT baseline.

