# OpenReview forum: "BRIDGE - Building Reinforcement-Learning Depth-to-Image Data Generation Engine for Monocular Depth Estimation"
_ICLR.cc/2026/Conference — ICLR 2026 Conference Desk Rejected Submission_

### Official Review · Reviewer_BMGy · 2025-10-26

**Soundness:** 3
**Presentation:** 3
**Contribution:** 3
**Rating:** 8
**Confidence:** 3

**Summary:**

The paper introduces BRIDGE for monocular depth estimation that pairs a reinforcement learning optimised depth-to-image generator with a hybrid supervision strategy. The generator produces approximately twenty million realistic RGB images directly from source depth maps in synthetic datasets (Hypersim, TartanAir), using reward-gradient training that balances depth consistency from depth inversion similarity with a CLIP based aesthetic score. This makes it so each RGB image is aligned with reliable depth. Training then fuses teacher pseudo labels with similarity guided masks based on ORB feature registration and SSIM to inject high precision ground truth depth where the generated RGB matches the original scene. The monocular depth model uses a DINOv2 Giant encoder and a DPT head with scale and shift invariant and gradient matching losses and a separate scale head for metric depth. In zero shot evaluations BRIDGE reports state of the art results on NYU v2, ScanNet, ETH three D, and Sintel, and competitive results on KITTI. Ablations attribute the gains to both the reinforcement learning depth to image data and the hybrid supervision.

**Strengths:**

- Monocular depth estimation is foundational in computer vision and this paper addresses a well-known bottleneck: scarcity and quality of depth-annotated data. By generating ~20mil RGB-D image pairs using their engine, the proposed method promises to resolve this major constraint. In doing so they claim improved generalisation across indoor/outdoor, zero-shot settings and better fine-detail capture.

- The paper first trains a depth-to-image generator via RL that translates source depth maps into realistic RGB images while preserving geometric structure (Sec. 2.1). Then it uses a hybrid supervision method with teacher model pseudo-labels for large scale and similarity-guided high precision ground truth for fine regions (Sec. 2.2). This is novel in that many prior works either rely purely on synthetic data or pseudo-labels but do not explicitly build a large RL-optimised generation engine that maps depth to image at this scale.

- The papers evaluates the method on five mainstream benchmarks—KITTI, NYUv2, ScanNet, ETH3D, Sintel in zero-shot settings and also for metric depth transfer. They show SOTA or near-SOTA results. For relative depth estimation they report the best performance on most of the datasets (Table 1) comparing to strong recent baselines like Depth Anything V2.

- The pipeline is clearly presented with synthetic teacher model plus D2I RL generator, similarity-masking and hybrid supervision and training a student depth model. The paepr gives enough implementation details (Sec. 3) to reproduce many elements. The structuring of contributions is explicit, which helps understand what was done at each step.

- The paper uses the DINOv2-Giant encoder together with a DPT head (Sec. 2.3) for depth estimation. In the proposed model selection ablation, the paper justifies the use of DINOv2-Giant for the generated large-scale data. This indicates performance due to a strong choice for architecture/representation and not purely data-scale.

- The D2I generator’s reward is explicitly designed to enforce geometrical accuracy (via depth map inversion similarity) and aesthetic quality. Additionally, they use registration (ORB features) followed by SSIM comparisons to mask and filter high-precision regions (Sec. 2.2). This focus on geometric alignment is important because many synthetic/generative data approaches have issues with domain or structural correctness.

**Weaknesses:**

- While BRIDGE improves empirical performance, the main novelty lies in using reinforcement learning to guide depth-to-image generation and combining pseudo-labels with high-precision masks. The final depth model relies on an existing DINOv2-Giant encoder and DPT head, so the conceptual advance is largely in pipeline engineering rather than a new principle for depth reasoning. This is not a major issue as the contributions are still useful.

- The paper briefly describes the reward structure of the depth-to-image generator but does not analyse how different reward terms contribute to the final RGB–D alignment or depth quality. There are no ablation experiments isolating the CLIP reward, the depth consistency term or the reinforcement learning optimiser itself. Essentially, it is unclear whether RL is essential or whether similar results could be achieved with standard diffusion or adversarial training.

**Questions:**

The paper mentions that the depth-to-image model is trained via a reward-gradient-driven direct optimisation process to ensure both geometric accuracy and visual realism. Could the authors clarify how this reinforcement learning component differs in practice from standard diffusion fine-tuning or reward-weighted loss formulations? Specifically, how important is the RL itself to achieving the reported geometric consistency and did the authors observe performance degradation if trained without the reward gradient?

---

> ### Author Response · Authors · 2025-11-21
>
> **Q1: Novelty Contribution.**
>
> **Data-Centric Paradigm:** We sincerely appreciate the reviewer's positive evaluation and recognition of the practical value of our work. We respectfully clarify that the core contribution of BRIDGE represents a paradigm shift from **"Model-Centric"** to **"Data-Centric"** AI. In the current landscape of Monocular Depth Estimation (MDE), the primary bottleneck is no longer model architecture (ViT, DPT), but the scarcity of high-quality, diverse, and geometrically consistent data. By demonstrating that an RL-driven data engine combined with Hybrid Supervision can significantly outperform specialized SOTA architectures using standard backbones, we offer a new, verifiable direction for the field: systematic data generation itself is a foundational algorithmic innovation.
>
> **Q2: Necessity of RL vs. Standard SFT and Reward Ablation.**
>
> **Ablation Setup:** we conducted an ablation study. We trained a Standard SFT version of the generator (using the same Flux backbone) and compared it with our RL-optimized versions (Depth Reward Only vs. Full). We then trained the downstream MDE model (DINOv2-Large) using 1M generated images from each variant.
>
> | Method | ScanNet:$\delta_1 \uparrow$ | ScanNet: AbsRel $\downarrow$ | ETH3D:$\delta_1 \uparrow$ | ETH3D: AbsRel $\downarrow$ |
> | :--- | :---: | :---: | :---: | :---: |
> | Standard SFT | 0.952 | 0.063 | 0.946 | 0.079 |
> | Ours (Depth Reward Only) | **0.960** | 0.062 | 0.950 | 0.077 |
> | **Ours (Full: Depth + Aesthetic)** | **0.960** | **0.061** | **0.952** | **0.074** |
>
> **Analysis 1: RL is Essential for Geometry (SFT vs. RL):**
>  As shown in the table above and the revised paper appendix fig 8, switching from Standard SFT to RL (Full) yields a significant performance jump ($\delta1$ on ScanNet improves from 0.952 to 0.960).
> * **Reasoning:** Standard SFT relies on noise prediction loss, which struggles to enforce strict pixel-level alignment between the generated image and the condition depth. This misalignment renders the "Hybrid Supervision" (mask-based fine-tuning) less effective because the generated "ground truth" regions are not sufficiently accurate.
> * **RL Advantage:** RL, via direct gradient feedback from the Depth Reward, forces the generator to prioritize geometric consistency above all else. This creates high-fidelity data that allows the model to fully leverage the potential of the hybrid supervision strategy.
>
> **Analysis 2: Role of Aesthetic Reward:**
>
> While the quantitative gain from adding the Aesthetic Reward appears marginal on ScanNet (though improving from 0.950 to 0.952 on ETH3D), its impact is primarily qualitative and semantic.
>
> **Qualitative Observation:** With "Depth Reward Only," the model occasionally generates objects that fit the geometry but violate semantic logic (generating a twisted car in a riverbed-shaped depression). The Aesthetic Reward corrects this by steering generation towards semantically plausible content (generating a boat instead). This semantic logicality significantly enhances the model's robustness in diverse "in-the-wild" scenarios.

---

### Official Review · Reviewer_Jp7h · 2025-10-30

**Soundness:** 3
**Presentation:** 3
**Contribution:** 2
**Rating:** 4
**Confidence:** 2

**Summary:**

The paper considered the oldest pain-point in MDE: lack of large-scale, high-quality RGB-D data. It turns the problem upside-down: instead of collecting more real RGB-D pairs, it trains an RL-tuned diffusion model that converts existing synthetic depth maps into 20 M photo-real RGB images whose geometry is provably consistent with the input depth. We can roughly say it is a data augmentation approach.
A hybrid supervision scheme then fuses (1) teacher-model pseudo-labels on the new RGB with (2) high-precision synthetic depth wherever the generated and original RGB are sufficiently similar (SSIM > 0.85). (3) A vanilla DINOv2-Giant + DPT head is finally trained on this 20 M set. The resulting model outperforms the Depth-Anything-V2 (62 M images) on every indoor benchmark and matches or surpasses the best metric-depth specialists on KITTI/NYU while using 3× fewer data. Extensive ablations, metric-depth fine-tuning and conditional-synthesis sanity checks are provided.

**Strengths:**

1) The use of RL to optimize the D2I generation process is novel and effective (according to the reported results). The RL formulation ensures that the generated images are not only visually realistic but also geometrically consistent with the source depth maps. This approach significantly mitigates common issues like geometric artefacts and structural distortions found in traditional D2I models.
2) The hybrid supervision strategy integrates teacher model pseudo-labels with high-precision ground truth depth. This dual supervision ensures that the model benefits from both the broad coverage of pseudo-labels and the precision of ground truth labels, leading to improved accuracy and robustness.
3) The use of similarity detection methods (SSIM and gradient analysis) to create masks for high-precision regions is reasonable. This approach ensures that the model receives reliable supervision during training, enhancing the overall quality and utilization efficiency of the labels.

**Weaknesses:**

1）The choice of SSIM threshold (0.85) and other hyperparameters for mask generation is arbitrary. A sensitivity analysis to show how these thresholds affect the final performance would be valuable.
2)  The paper claims to generate a diverse dataset, but it does not provide quantitative metrics or a valid discussion to support this claim. We believe that the scale of data augmentation will definitely generate more diversity but is it too costly? We spent too much but earn very little. What is the contribution of this 'diversity' to the final performance? More discussions are needed.
3) The paper uses a fixed MiDaS-small model for depth inversion in the RL reward function. This method may introduce biases if the MiDaS model fails on certain types of images. Exploring alternative depth inversion methods or robustness checks would be beneficial.

**Questions:**

1) Following the weakness discussed above, what are the limitations of using MiDaS-small for depth inversion in the RL reward function?
2) The model's performance on the KITTI dataset is not optimal. Any investigation into these issues?

---

> ### Author Response · Authors · 2025-11-21
>
> **Q1: SSIM Threshold Choices.**
>
> **Rationale:** We appreciate the reviewer's suggestion. We clarify that our initial choice of 0.85 was empirically determined to strike a balance between Geometric Fidelity (which requires a high threshold) and the Pixel Density needed for effective fine-tuning (which requires a lower threshold).
>
> **Sensitivity Analysis:** To quantify this, we conducted a sensitivity analysis on the SSIM threshold while keeping other hyperparameters (3x3 erosion, 518 crop size) fixed. As shown in the table below, performance is robust within the [0.80, 0.85] range.
>
> | Method | KITTI: $\delta_1 \uparrow$ | KITTI: AbsRel $\downarrow$ | NYUv2: $\delta_1 \uparrow$ | NYUv2: AbsRel $\downarrow$ | ScanNet: $\delta_1 \uparrow$ | ScanNet: AbsRel $\downarrow$ |
> | :--- | :---: | :---: | :---: | :---: | :---: | :---: |
> | Pseudo-labels Only | 0.924 | 0.099 | 0.958 | 0.069 | 0.957 | 0.069 |
> | SSIM 0.70 | 0.922 | 0.099 | 0.958 | 0.070 | 0.955 | 0.072 |
> | SSIM 0.80 | **0.926** | 0.099 | 0.959 | 0.068 | 0.958 | 0.062 |
> | **SSIM 0.85 (Ours)** | **0.926** | **0.098** | **0.960** | **0.067** | **0.960** | **0.061** |
> | SSIM 0.90 | 0.925 | **0.098** | **0.960** | 0.069 | **0.960** | 0.064 |
>
> As demonstrated by our ablation studies, we observed the following:
> * When the threshold is set too low (≤ 0.8), geometrically inconsistent pixels may be included in the mask, potentially introducing slight noise or interference during training.
> * Conversely, when the threshold is set too high (≥ 0.9), the resulting mask becomes overly sparse. This reduces the number of available pixels for effective model fine-tuning.
>
> In summary, the model's performance is relatively stable for SSIM values between 0.80 and 0.85. Thresholds below 0.8 or above 0.9 provide diminished benefits, and can even lead to performance degradation due to either excessive noise in the masked regions or excessive sparsity.

---

> ### Author Response · Authors · 2025-11-21
>
> **Q2: Quantification of Diversity and Cost-Benefit Analysis.**
>
> * **High Return on Investment:** We believe the computational "cost" of generation yields a significantly high "return." While generating images incurs overhead, it drastically reduces the data scale required for SOTA performance. BRIDGE achieves superior results using only 20M generated images, outperforming competitors like Depth Anything V2 which relies on 62M real-world images.
> * **Contribution of Diversity:** We compared performance across different generation ratios (K images per depth map) using a DINOv2-Large backbone. As shown in the table below:
>
> | Generate Pair | KITTI: $\delta1\uparrow$ | KITTI: $AbsRel\downarrow$ | NYUv2: $\delta1\uparrow$ | NYUv2: $AbsRel\downarrow$ | Scannet: $\delta1\uparrow$ | Scannet: $AbsRel\downarrow$ | ETH3D: $\delta1\uparrow$ | ETH3D: $AbsRel\downarrow$ | Sintel: $\delta1\uparrow$ | Sintel: $AbsRel\downarrow$ |
> | :--- | :---: | :---: | :---: | :---: | :---: | :---: | :---: | :---: | :---: | :---: |
> | Baseline (K=0, 500K) | 0.921 | 0.097 | 0.957 | 0.063 | 0.955 | 0.060 | 0.962 | 0.063 | 0.590 | 0.785 |
> | K=1, 500K | 0.919 | 0.099 | 0.955 | 0.064 | 0.950 | 0.064 | 0.960 | 0.066 | 0.584 | 0.789 |
> | K=4, 2M | 0.931 | 0.090 | 0.964 | 0.054 | 0.969 | 0.046 | 0.971 | 0.058 | 0.615 | 0.687 |
> | **K=free, 20M** | **0.937** | **0.081** | **0.978** | **0.045** | **0.972** | **0.049** | **0.989** | **0.032** | **0.688** | **0.588** |
>
> As K increases, all metrics consistently improve ($\delta1$ on ETH3D improves from 0.960 to 0.989). This empirically proves that the "diversity" provided by our engine serves as a crucial performance driver.

---

> ### Author Response · Authors · 2025-11-21
>
> **Q3: Clarification on RL Reward Model.**
>
> * **Correction:** We wish to clarify a misunderstanding. We did not use MiDaS-small for the RL reward function.
> * **Actual Mechanism:** Instead, we utilized the Teacher Model described in our methodology(chapter 2.1) (pretrained on 1M synthetic data). The reward is calculated as the Cosine Similarity between the pseudo-depth of the generated RGB (predicted by the Teacher) and the Ground Truth (GT) depth. This ensures the generated images maintain structural alignment with the source geometry. Since the teacher is trained on domain-aligned data, this approach avoids the potential bias or failure modes associated with using a fixed external model like MiDaS-small.
>
> **Q4: Performance Analysis of KITTI.**
>
> The limited gains of BRIDGE on KITTI primarily stem from a structural mismatch between the target task and the available supervision. The LiDAR-based ground truth of KITTI is both sparse and unevenly distributed, failing to reflect the model's improvements in large, unsupervised regions. Furthermore, KITTI's metrics are biased towards mid-to-short range depths in regular driving scenarios, making them insensitive to BRIDGE's advantages in capturing fine-grained details.
> Therefore, we argue that this limitation arises from the evaluation framework itself, rather than a fundamental deficiency in the model's capabilities. Future work could address this by generating D2I data that simulates LiDAR sampling patterns.

---

### Official Review · Reviewer_tqP2 · 2025-10-31

**Soundness:** 2
**Presentation:** 2
**Contribution:** 2
**Rating:** 4
**Confidence:** 3

**Summary:**

This paper designs a pipeline for training depth estimation model together with synthesizing images from depth map. The pipeline contains several stages: image generation from depth maps, pseudo label generation, model training. The experiments show that the trained model achieves SOTA performance in comparison with previous methods.

**Strengths:**

- The data synthesis for depth estimation model training is useful.

- The performance is good.

- The paper is well-written.

**Weaknesses:**

- The main contribution is providing a data synthesis pipeline for depth estimation model training, which has very limited novelty.

- The depth-to-image strategy for data synthesis is used before [1] that is not been discussed.

- Given a depth map, the generated RGB image has a very similar structure. Does this lead to data redundancy? In other words, for a fixed number of depth maps, is it always better to generate more RGB images? Could the data redundancy from generating too many images lead to limited performance gains or even overfitting?

- How the proposed  mask generation strategy impacts the performance.


[1] Zhang et al., Atlantis: Enabling Underwater Depth Estimation with Stable Diffusion.

**Questions:**

see Weaknesses

---

> ### Author Response · Authors · 2025-11-21
>
> **Q1: Novelty Contribution.**
>
> **Systematic Innovation:** Thank you for your valuable feedback. We would like to clarify that while data synthesis is indeed a core component of our work, our approach differs from previous methods that rely on passive data collection or fixed generation rules. Our key innovation lies in the introduction of a **Depth-to-Image (D2I) engine optimized by Reinforcement Learning (RL).** This active generation strategy enables us to dynamically create geometrically consistent and content-diverse data tailored to the model's training needs, effectively mitigating the data scarcity issue in specific scenarios found in existing datasets.
>
> Furthermore, we do not just generate data; we integrate this process with **a mixed-supervision training strategy for Monocular Depth Estimation (MDE).** The synergy between these two components collectively enhances the generalization ability of the MDE model. We believe that this combination of "active data generation + a tailored training strategy" offers a novel and effective pathway for improving the performance of general-purpose depth estimation models.
>
> **Q2: Related Work.**
>
> **Comparison with Atlantis:** We thank the reviewer for mentioning the awesome work: Atlantis. We have added a discussion of it in the revised version of our paper. While both methods utilize the D2I concept, their objectives and mechanisms differ fundamentally. Atlantis relies on triplet supervision for style transfer, specifically targeting underwater scenes. In contrast, BRIDGE employs a closed-loop RL optimization to maximize geometric consistency across diverse indoor/outdoor domains, ensuring the generated RGB images align pixel-perfectly with the source depth maps.

---

> ### Author Response · Authors · 2025-11-21
>
> **Q3: Data Redundancy and Overfitting (Is more always better?).**
>
> * **Geometry vs. Appearance:** We agree that, theoretically, infinite generation could lead to redundancy. However, in MDE, geometric structure serves as the "signal," while appearance often acts as a "nuisance variable." Traditional "one-to-one" datasets risk inducing the model to learn spurious correlations between specific textures and depth [1, 2]. By generating multiple RGB variations for a single depth map, we force the model to learn the invariant geometric structure rather than memorizing textures.
> * **Benefits Outweigh Redundancy:** To empirically address the concern "is generating more images always better?", we compared performance under different generation ratios (K images per depth map) using a DINOv2-Large backbone.
>
> | Generate Pair | KITTI: $\delta1\uparrow$ | KITTI: $AbsRel\downarrow$ | NYUv2: $\delta1\uparrow$ | NYUv2: $AbsRel\downarrow$ | Scannet: $\delta1\uparrow$ | Scannet: $AbsRel\downarrow$ | ETH3D: $\delta1\uparrow$ | ETH3D: $AbsRel\downarrow$ | Sintel: $\delta1\uparrow$ | Sintel: $AbsRel\downarrow$ |
> | :--- | :---: | :---: | :---: | :---: | :---: | :---: | :---: | :---: | :---: | :---: |
> | Baseline (K=0, 500K) | 0.921 | 0.097 | 0.957 | 0.063 | 0.955 | 0.060 | 0.962 | 0.063 | 0.590 | 0.785 |
> | K=1, 500K | 0.919 | 0.099 | 0.955 | 0.064 | 0.950 | 0.064 | 0.960 | 0.066 | 0.584 | 0.789 |
> | K=4, 2M | 0.931 | 0.090 | 0.964 | 0.054 | 0.969 | 0.046 | 0.971 | 0.058 | 0.615 | 0.687 |
> | **K=free, 20M** | **0.937** | **0.081** | **0.978** | **0.045** | **0.972** | **0.049** | **0.989** | **0.032** | **0.688** | **0.588** |
>
> The results demonstrate that as K increases (up to 20M), model performance consistently improves without signs of overfitting. This confirms that at the current scale, our **"One-Geometry-Many-Appearances"** strategy effectively enhances model robustness by decoupling appearance from geometry. We will continue to explore this issue in more detail in the future.
>
> **Q4: Impact of Mask Generation Strategy.**
>
> **Precision Calibration:** Our mask strategy complements the large-scale generation by acting as a "Precision Calibrator." While pseudo-labels provide broad supervision, they inevitably contain noise. Our mask generation, based on SSIM and ORB feature matching, automatically identifies regions where the generated image is highly consistent with the source geometry. In these verified regions, we enforce exact Ground Truth depth supervision. As validated in Table 3 of the main paper, this hybrid approach significantly improves the model's ability to capture fine-grained details compared to using pseudo-labels alone.
>
> ---
>  [1] Zhao H, et al. DME: Unveiling the Bias for Better Generalized Monocular Depth Estimation.
>
>  [2] Wang L, et al. DID-M3D: Decoupling Instance Depth for Monocular 3D Object Detection.

---

### Official Review · Reviewer_oh7b · 2025-10-31

**Soundness:** 3
**Presentation:** 3
**Contribution:** 3
**Rating:** 8
**Confidence:** 4

**Summary:**

This paper introduces BRIDGE, a framework for MDE that tackles the challenges of data scarcity and quality limitations. It achieves this by leveraging a Reinforcement Learning (RL)-optimized Depth-to-Image (D2I) generation engine. BRIDGE is capable of synthesizing over 20 million highly realistic and geometrically accurate RGB images, each paired with its ground truth depth, all derived from diverse source depth maps. A pivotal contribution lies in its hybrid supervision strategy, which combines high-precision ground truth depth (meticulously filtered by similarity detection) with pseudo-labels generated by a teacher model. The authors demonstrate that training an MDE model on this generated dataset, utilizing their hybrid supervision, allows BRIDGE to achieve state-of-the-art performance across various benchmarks (NYUv2, ScanNet, ETH3D, Sintel, DA2K). Remarkably, it does so while using significantly less training data than some existing formidable methods. The paper particularly highlights BRIDGE's exceptional capability in capturing fine-grained details, maintaining robustness in complex scenes, and exhibiting strong zero-shot generalization.

**Strengths:**

Novel Data Generation Paradigm: The RL-optimized Depth-to-Image (D2I) generation engine is a highly original and effective solution to data scarcity. By using RL to optimize for both visual realism and geometric consistency (via depth loss and aesthetic reward), the method generates a massive and high-quality synthetic RGB-D dataset.

Effective Hybrid Supervision Strategy: The proposed hybrid supervision, combining teacher pseudo-labels for broad coverage and similarity-guided high-precision ground truth for fine-tuning, is a strong methodological contribution. It intelligently balances the quantity provided by pseudo-labeling with the quality of true annotations, leading to more robust and accurate models.

 Robustness and Generalization: The model demonstrates excellent zero-shot generalization to diverse and complex "in-the-wild" scenes, including challenging elements like reflective surfaces and transparent objects. This is crucial for practical MDE applications.

**Weaknesses:**

Training Details for RL-D2I: While the RL-D2I generation engine is the core innovation, the paper's detailed description of its training process is somewhat lacking. For example, a deeper elaboration could be provided, such as mentioning the training duration, an overview of the GPU resources used, and the final improvement in reward results.

Specifics of the Scale Head: The paper mentions a specialized scale head for zero-shot metric depth estimation, stating that it is decoupled and trained separately. To facilitate a more comprehensive understanding of the metric depth estimation capabilities, it would be very helpful if further details regarding the architecture of this scale head and its specific training procedure could be provided.

**Questions:**

Supplementary Notes on RL-D2I Model Training: Could authors supplement the approximate training duration, the type and number of GPU resources used, and the changes in reward results for the RL-optimized D2I model training?

Scale Head Architecture and Training: Could the authors provide a brief description of the architecture of the specialized scale head? Furthermore, since it's trained separately, what is the specific loss function used for its training, and on which specific dataset(s) is it optimized?

---

> ### Author Response · Authors · 2025-11-21
>
> **Q1: Implementation Details of RL-D2I.**
>
> **RL-D2I Training Dynamics:** We appreciate the reviewer's inquiry regarding the specific training configurations. To ensure optimization stability, the RL-D2I model was trained using 64 NVIDIA A800 (80GB) GPUs with 128 batch sizes. The entire RL optimization phase is training 1,500 steps. During this process, the reward function demonstrated significant convergence, improving from an initial value of **5.0683** to **6.5884**, indicating that the geometric consistency of the generated data was effectively enhanced.
>
> | Step | Mean Reward |
> | :---: | :---: |
> | $0$ | $5.0683$ |
> | $200$ | $5.2497$ |
> | $700$ | $5.9673$ |
> | $1200$ | $6.2833$ |
> | $1500$ | $6.5884$ |
>
>
> **Q2: Scale Head Structure Details.**
>
> **Scale Head Architecture & Training:** Following the design of MoGe-2 [1], our specialized Scale Head is a lightweight, task-specific decoder that operates independently of the depth estimation head.
>  * **Architecture Design:** It consists of a two-layer MLP that takes the CLS token feature from the DINOv2 encoder as input. The CLS token effectively captures the global, contextualized features of the image, which are essential for predicting a global scalar value.
> * **Loss Function:** Since scale prediction is a scalar regression task, we employed the standard $L_1$ Loss to ensure the model accurately regresses the global scaling factor s required to convert relative depth into absolute metric depth. The loss function $L_{scale}$  is defined as:
>
> $$\space \space \space \space \space \space  L_{\text{scale}} = \frac{1}{N} \sum_{i=1}^{N} |s_{i}^{\text{pred}} - s_{i}^{\text{gt}}|$$
>
> where  $s_{i}^{\text{pred}}$ and $\space s_i^{gt}$ represent the predicted and ground truth scale factors, respectively. By training on this objective, we ensure the Scale Head can robustly recover the metric scale.
>
> ---
> [1] Wang R, Xu S, Dong Y, et al. MoGe-2: Accurate Monocular Geometry with Metric Scale and Sharp Details[J]. arXiv preprint arXiv:2507.02546, 2025.

---

> ### Comment · Reviewer_oh7b · 2025-11-26
> **Official comments of Reviewer oh7b**
>
> Thanks for the detailed reply to my concerns. I maintain my recommendation for accepting this paper.

---

### Author Response · Authors · 2025-11-21

We sincerely thank reviewers oh7b, tqP2, Jp7h, and BMGy for their time and effort in reviewing our paper and providing insightful feedback. We are grateful for the positive comments on our novel data generation paradigm (oh7b, Jp7h, BMGy), strong performance (oh7b, tqP2), and robust generalization (oh7b). Your comments have been instrumental in improving the quality and completeness of our work.
Regarding the reviewers' common concerns about the novelty in relation to prior work, the specifics of our RL-D2I training, and the cost-effectiveness of our data generation strategy, we have provided extensive clarifications and new experimental results. We have added a detailed comparison with related works and a new ablation study to demonstrate the performance gains from our data generation pipeline.
We have uploaded a revised draft of our paper. The main revisions are summarized as follows:
* **Expanded the related works section to include a detailed comparison with Atlantis [1]**, clarifying the key differences and highlighting the novelty of our RL-based approach.
* **Added a new visualization comparing Depth SFT and RL-based optimization** to intuitively demonstrate the superiority of our RL-based approach in maintaining geometric consistency.
We sincerely value your reviews and believe they have greatly contributed to making the new revision more comprehensive and solid. We have also posted individual responses to each of your comments below.

[1] Zhang et al., Atlantis: Enabling Underwater Depth Estimation with Stable Diffusion.

---

### Note · Program_Chairs · 2026-01-02
**Submission Desk Rejected by Program Chairs**

Abstract breaks anonymity with a GitHub link.